# Q1Fold: A Qubit-Efficient Hybrid Quantum-Classical Convolutional Neural Network for RNA secondary Structure Prediction

## Abstract

RNA 2D structure prediction remains a critical challenge in computational biology, with existing thermodynamic and deep learning approaches facing limitations in modeling complex interactions and data requirements. We introduce Q1Fold, a hybrid quantum-classical convolutional network for RNA secondary structure prediction. The model integrates a compact variational quantum circuit with a classical 2D ResNet architecture, where the quantum circuit generates expressive features from local sequence windows using minimal qubits. This design avoids barren plateaus and is compatible with current Noisy Intermediate-Scale Quantum Computers. Despite using significantly fewer parameters, Q1Fold achieves competitive performance on standard benchmarks compared to state-of-the-art methods. The extracted quantum features also demonstrate superior capability in representing local structural motifs such as hairpins. Our work establishes a practical route toward quantum-enhanced computational RNA biology.

## 1 Introduction

Ribonucleic acid (RNA) plays fundamental roles in numerous cellular processes, including protein synthesis, gene regulation, catalytic reactions, and sensing of molecular signals (Sahin et al., 2014; Pardi et al., 2018). The secondary (2D) structure of RNA is crucial for understanding its biological function and serves as a foundation for tertiary structure formation (Tinoco Jr & Bustamante, 1999; Budnik et al., 2023). The accurate prediction of RNA 2D structure from primary sequence alone remains one of the most important and challenging problems in computational biology, with significant implications for drug discovery, vaccine development, and synthetic biology applications (Chaudhary et al., 2021). Traditional approaches to RNA 2D structure prediction have relied primarily on thermodynamic models that minimize free energy based on nearest-neighbor parameters (Zuker, 2003). Methods such as ViennaRNA (Lorenz et al., 2011) and MXfold2 (Sato et al., 2021) have achieved considerable success, but face inherent limitations when dealing with complex structural motifs, particularly pseudoknots and long-range base pairs (Lyngsø & Pedersen, 2000). Early machine learning models marked a significant shift in RNA structure prediction by learning directly from data, bypassing explicit energy calculations (Singh et al., 2019). Methods like E2Efold (Chen et al., 2020), UFold (Fu et al., 2022) and sincFold (Bugnon et al., 2024) demonstrated that deep learning architectures, particularly convolutional neural networks, could successfully model spatial relationships between nucleotides. Despite their improvements over thermodynamic approaches, these models remain limited by their sequence embedding strategies and feature representation capabilities, often struggling to capture the full complexity of RNA structural patterns and long-range dependencies (Chen et al., 2022). More recently, the field has witnessed the emergence of large language model (LLM) and foundation model (FM) based approaches that leverage pre-trained sequence representations to enhance RNA structure prediction (Wu et al., 2025). Models such as RNAErnie (Wang et al., 2023) and Depfold (WANG & Cohen, 2025) utilize transformer architectures and self-attention mechanisms to capture complex sequence-structure relationships. Although these models have shown promising results, their enormous parameter requirements, often in the hundreds of millions to billions, pose a significant risk of overfitting given the currently limited size of available RNA structure datasets, representing a fundamental bottleneck in achieving robust and generalizable predictions of RNA structure (Wu et al., 2025). Quantum computing offers a fundamentally different pathway for addressing these computational challenges through entanglement

and superposition (Biamonte et al., 2017). The exponential scaling of quantum Hilbert spaces and the ability of quantum circuits to generate complex entangled states present unique advantages for capturing intricate patterns in RNA folding (Fox et al., 2022). However, purely quantum-based RNA folding models remain in their infancy, with existing implementations suffering from issues such as barren plateaus, limited qubit availability, and performance that have yet to match state-of-the-art (SOTA) machine learning models (Alevras et al., 2024; Kumar et al., 2025).

In this work, we introduce Q1Fold, a novel hybrid quantum-classical convolutional neural network (HQC-CNN) that addresses the limitations of both classical and quantum approaches to RNA 2D structure prediction. Our method leverages quantum convolutional layers for enhanced feature extraction while maintaining compatibility with current Noisy Intermediate-Scale Quantum (NISQ) devices. By integrating quantum convolution with classical CNN architectures, Q1Fold achieves competitive performance while requiring only a small number of qubits proportional to the local window size rather than the full sequence length. The contributions of our work are summarized below:

- We propose Q1Fold, the first HQC-CNN model specifically designed for RNA 2D structure prediction. Unlike existing quantum approaches that primarily use Quantum Approximate Optimization Algorithm (QAOA) or Variational Quantum Eigensolver (VQE) frameworks, Q1Fold integrates quantum circuits directly into CNN architectures, achieving significant parameter reduction while maintaining competitive performance compared to SOTA classical methods.
- We design a qubit-efficient variational quantum circuit (VQC) with learnable data embedding for 1D feature extraction of RNA sequences. McClean et al. (2018) demonstrated barren plateau effects scale exponentially with the number of qubits, our qubit-efficient design circumvents this fundamental limitation.
- We introduce a position-, context- and balance-aware quantum entanglement scheme that encodes positional, C/G vs A/U contextual and purine-pyrimidine balanced information in dedicated qubits, producing quantum features with significantly enhanced expressivity. This quantum feature richness enables accelerated learning dynamics, achieving improved validation in early epochs and competitive overall performances.

## 2 BACKGROUND

### 2.1 RNA 2D STRUCTURE PREDICTION

RNA 2D structure prediction aims to determine the pattern of base pairings in an RNA molecule given only its primary sequence. The 2D structure can be represented as a contact matrix where entry (i,j) indicates whether nucleotides at positions i and j form a base pair. Valid 2D structures must satisfy several biological constraints: only Watson-Crick (A-U, G-C) and wobble (G-U) base pairs are allowed, no sharp loops with fewer than 4 unpaired nucleotides can form, and each nucleotide can participate in at most one base pair (Huang et al., 2019).The computational complexity of this problem stems from the exponential search space of possible base-pairing configurations for sequences of length n, making exhaustive enumeration intractable (Tinoco Jr & Bustamante, 1999). Additionally, the presence of pseudoknots—non-nested base pairs that cross each other—significantly increases the difficulty, as standard dynamic programming approaches cannot handle these structures efficiently (Lyngsø & Pedersen, 2000). The limited availability of experimentally validated RNA structures further constrains the development and evaluation of prediction methods, with existing databases containing only tens of thousand of high-quality annotations compared to the millions of known RNA sequences.

### 2.2 HYBRID QUANTUM-CLASSICAL CNN

Hybrid quantum-classical convolutional neural networks (HQC-CNNs) represent an emerging paradigm that combines quantum circuit feature extraction with classical neural network scalability (Henderson et al., 2020; Cong et al., 2019). The core innovation lies in replacing classical convolutional filters with parametric quantum circuits that process data in exponentially large Hilbert spaces, exploiting superposition and entanglement to generate complex, non-linear feature representations (Liu et al., 2021). HQC-CNNs have demonstrated competitive performance on image classification benchmarks like MNIST and Fashion-MNIST using significantly fewer parameters than

classical CNNs (Henderson et al., 2020). More recently, quantum convolution has shown particular promise for time series data, where quantum circuits' ability to capture temporal correlations through entanglement aligns well with long-range dependencies and complex periodic patterns (Orka et al., 2025). The study by Orka et al. demonstrated that fully quanvolutional networks could outperform classical methods on 20 UEA and UCR time series datasets while using 6.5 times fewer parameters on average, suggesting quantum effects provide representational advantages even in the NISQ era. The application to biological sequences presents unique opportunities, as the exponential feature space of quantum circuits could capture combinatorial sequence-structure relationships more efficiently than classical approaches (Chen et al., 2023). The parameter efficiency of quantum circuits directly addresses RNA structure prediction's fundamental challenge: the scarcity of high-quality training data relative to model complexity, potentially reducing overfitting risks compared to foundation models with hundreds of millions of parameters.

## 3 Q1FOLD MODEL

As illustrated in Figure 1(a), our model consists of two main modules: the 1D feature extraction module and the 2D pairing prediction module. First of all, the input RNA sequence of length L is one hot encoded. Then it is processed by our quantum 1D (Q1D) feature extraction layer, which employs VQC for feature representation. The resulting 1D features are then processed with two 1D convolutional layers and matrix product operations to generate a 2D pairing probability map. This 2D map is subsequently processed through a classical 2D ResNet CNN network with postprocessing to produce the final RNA contact map prediction.

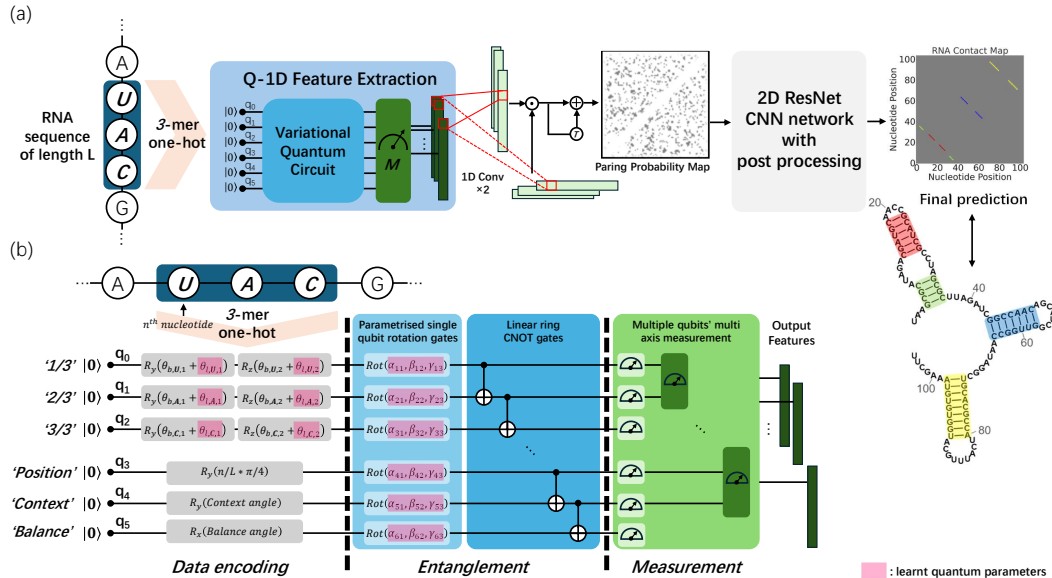

Figure 1: Q1Fold model architecture. (a) Overall pipeline: RNA sequence (length L) is one-hot encoded, processed through a Q1D feature extraction layer using 3-mer sliding windows, followed by 1D convolutions and matrix products to generate a 2D pairing probability map. A 2D ResNet CNN with post-processing produces the final L×L contact map. (b) 6-qubit variational quantum circuit with three stages: (i) Data encoding using learnable rotation gates for nucleotide triplets ($q_0$-$q_2$) and biology-aware qubits for position ($q_3$), context ($q_4$), and balance ($q_5$); (ii) Entanglement via linear ring CNOT gates and parametrized rotations; (iii) Multi-axis measurements to extract quantum features.

## 3.1 Quantum 1D feature extraction circuit

As illustrated in Figure 1(b), the Q1D feature extraction layer implements a 6-qubit VQC that processes 3-nucleotide sliding windows from RNA sequences. The overall circuit architecture consists of three distinct stages: data encoding, parametrized entanglement, and measurement.

**Data Encoding Stage** The encoding stage maps classical RNA sequence information into quantum states using a 6-qubit scheme. The first three qubits ($q_0$, $q_1$, $q_2$) encode individual nucleotides using biochemically motivated base angles (A=[0,0], G=[$\pi$/2,0], C=[0,$\pi$/2], U=[$\pi$/2,$\pi$/2]) augmented with learned angles as rotation angles for RY and RZ gates. Qubit $q_3$ encodes positional information through position dependent rotations normalized to (0, $\pi$/4). The qubit $q_4$ encodes the sequence context, within the 3-mer window, via the context angle, which were calculated with energy-aware weighted nucleotide composition. The balance qubit $q_5$ encodes the purine-pyrimidine balance by calculating the angle dependent on the A/G to C/U ratio.

**Parametrized Entanglement Stage** The entanglement stage creates quantum correlations between encoded qubits through parametrized variational structure. After data encoding, parametrized Rot($\alpha$, $\beta$, $\gamma$) gates are applied to all qubits with learnable parameters $\boldsymbol{\theta} \in \mathbb{R}^{1 \times 6 \times 3}$ followed with linear ring entanglement through CNOT gates between adjacent qubits. This hierarchical entanglement structure enables the circuit to model complicated local nucleotide characteristics and enables high expressivity of the quantum circuit.

**Measurement stage** The measurement stage extracts features through a comprehensive set of quantum observables. Single-qubit measurements apply Pauli-X, Y, and Z operators to all six qubits, yielding 18 expectation values that capture individual qubit states. Two-qubit correlation measurements employ tensor products of Pauli operators (ZZ, XX, YY) on 11 strategically selected qubit pairs, producing 33 correlation features that encode pairwise quantum relationships. A single three-qubit measurement ZZZ on the nucleotide qubits ($q_0$ to $q_2$) captures higher-order correlations within the 3-mer window. As the result, we produced 52 channels of Q1D features for the subsequent layers via informative quantum measurements.

Overall, the 6-qubit design maintains a minimal qubit count that scales with the local window size rather than sequence length, effectively circumventing the barren plateau problem that severely limits larger VQCs. Specifically, due to the exponential suppression of gradients in variational quantum algorithms scales with the number of qubits McClean et al. (2018), our qubit-efficient approach is crucial for practical trainability. Combined with the biochemically-informed initialization with learnable embedding weights, this provides a good starting point for training, while preserving biology intuition about nucleotide properties.

## 3.2 Classical 2D ResNet CNN network with postprocessing

Following Q1D feature extraction, the model employs a classical deep learning architecture similar to sincFold (Bugnon et al., 2024) to refine and produce final RNA contact predictions.

**2D Map Construction** The 52-channel Q1D features undergo dimensionality reduction through two parallel 1D convolutional layers with rank $r = 64$. These are combined via outer product and symmetrized to enforce bidirectional base pairing: $Y_{final} = (Y + Y^T)/2$, generating an initial $L \times L$ pairing probability matrix.

**2D ResNet Architecture** The refinement network consists of an initial $7 \times 7$ convolutional layer (256 filters) followed by two ResNet blocks with bottleneck architectures. Each block implements: BatchNorm2D $\rightarrow$ ReLU $\rightarrow$ Conv2D (bottleneck) $\rightarrow$ BatchNorm2D $\rightarrow$ ReLU $\rightarrow$ Conv2D (expansion) with skip connections. The blocks use 256 and 128 bottleneck channels respectively, employing $5 \times 5$ kernels with dilation factor 3 to capture long-range dependencies. A final $5 \times 5$ convolutional layer produces the single-channel contact map.

**Loss Function and Training** The model employs a multi-component loss function:

$$\mathcal{L}_{total} = \mathcal{L}_{CE} + \beta\mathcal{L}_{aux} + \lambda_1\mathcal{L}_{L1} + \lambda_2\mathcal{L}_{reg} \tag{1}$$

where $\mathcal{L}_{CE}$ is the cross-entropy loss of the final prediction, $\mathcal{L}_{aux}$ is an auxiliary loss from the intermediate 2D map with $\beta = 0.15$, $\mathcal{L}_{L1}$ encourages sparsity with $\lambda_1 = 0.0005$, and $\mathcal{L}_{reg}$ is adaptive L2 regularization on quantum parameters. Separate optimizers are used: AdamW for quantum

parameters ($2 \times 10^{-4}$) and Adam for classical parameters ($1 \times 10^{-4}$), with ReduceLROnPlateau scheduling.

**Postprocessing** Raw predictions undergo postprocessing to ensure biological validity: canonical base pairing enforcement, symmetrization, binarization (threshold 0.5), minimum hairpin loop constraints (3 nt), and conflict resolution for overlapping pairs. This pipeline improves F1 scores by 2–3% on benchmark datasets.

The combination of quantum feature extraction with classical deep learning refinement enables Q1Fold to capture both local sequence patterns through quantum entanglement and global structural constraints through the 2D ResNet architecture, achieving competitive performance while maintaining parameter efficiency.

# 4 EXPERIMENTAL SETUP

## 4.1 DATASET

We evaluated Q1Fold on four well-known RNA structure prediction benchmark datasets, as well as a hairpin dataset built by ourselves.

**RNAStrAlign** (Tan et al., 2017) contains 37,149 RNA sequences from 8 RNA families. Following E2Efold (Chen et al., 2020) and MXfold2 (Sato et al., 2021), we first filter out the redundancies, retaining 30,879 unique structures. After that, we further filter away sequences longer than 512 nucleotides due to hardware limitation, leaving 19,966 sequences for experiments in our work.

**ArchiveII** (Saman Booy et al., 2022) contains 3,975 sequences from 10 families. After removing duplicates and limiting the sequence length to 512, we retained 3,380 sequences from 9 families. This data set serves as a test benchmark for RNA folding after training with the RNAStrAlign training split. Both RNAStrAlign and ArchiveII includes pseudoknots.

**bpRNA-1m** (Singh et al., 2019) contains 102,318 structures from 2,588 RNA families. Following SPOT-RNA (Singh et al., 2019), we use CD-HIT program (Fu et al., 2012) to remove similar sequences with a cut-off of 80%. For the remaining sequences, we follow the same partition between train, validation and test data that was proposed in SPOT-RNA. The data set was split to TR0 for training, VL0 for validation, TS0 for testing.

**bpRNA-new** (Sato et al., 2021) contains 1,500 structures derived from Rfam 14.4. It is used to assess cross-family generalization after training with TR0. Both bpRNA-1m and bpRNA-new are shorter than 500 nucleotide lengths and free of pseudoknots.

**Hairpin dataset** We extracted RNA hairpin motifs from 2001 sequences in the RNAStrAlign test subset using a simple pipeline. The extraction algorithm first identifies hairpin structures from dot-bracket notation through bracket matching, then validates candidates based on structural constraints: minimum stem length (3 bp), loop length (3–15 nt) and loop unpaired ratio (70%). In addition, thermodynamic stability score or hairpin energy ($\Delta G$) is calculated using the Turner nearest-neighbor energy model, in-

Table 1: Summary of datasets used.

| Dataset | Subset | #Seq. | Len. Range |
|---------|--------|-------|------------|
| RNAStrAlign | Train | 15,988 | 31–512 |
|  | Val | 1,977 | 33–511 |
|  | Test | 2,001 | 30–510 |
| ArchiveII | - | 3,380 | 28–512 |
| bpRNA-1m | TR0 | 10,814 | 33–498 |
|  | VL0 | 1,300 | 33–497 |
|  | TS0 | 1,305 | 22–499 |
| bpRNA-new | - | 1,500 | 33–489 |
| Hairpin | - | 5,354 | 11–45 |

corporating base pair formation energies, stacking interactions, terminal AU penalties, and loop initiation costs. Overlapping hairpins are resolved by retaining structures with the longest stems or the lowest $\Delta G$. This process yielded 5,349 unique hairpins with comprehensive annotations including position, sequence, structure, and stability scores (-86.1 to +5.9 kcal/mol), providing a biologically relevant dataset for evaluating RNA feature extraction methods. Table 1 summarizes the details of all the datasets used in our experiments.

## 4.2 MEASURES AND BASELINE MODELS

Following standard practice in RNA 2D structure prediction, we evaluate model performance using precision, recall, and F1 score metrics based on the correct identification of base pairs compared to ground truth structures. These metrics are widely adopted in the field and allow for direct comparison with existing methods.

We compare our proposed Q1Fold with several baseline methods, including: Energy based: RNAFold from viennaRNA (Lorenz et al., 2011), MXfold2 (Sato et al., 2021). Early learning based: E2Efold (Chen et al., 2020), UFold (Fu et al., 2022), sincFold (Bugnon et al., 2024). Recent large language and foundation model based: RNAErnie (Wang et al., 2023), DEPfold (WANG & Cohen, 2025).

## 5 RESULTS

### 5.1 QUANTUM FEATURE ANALYSIS FOR HAIRPIN LOCAL MOTIF

To examine the capability of quantum feature in capturing local RNA motifs, specifically hairpin, we designed two downstream tasks.

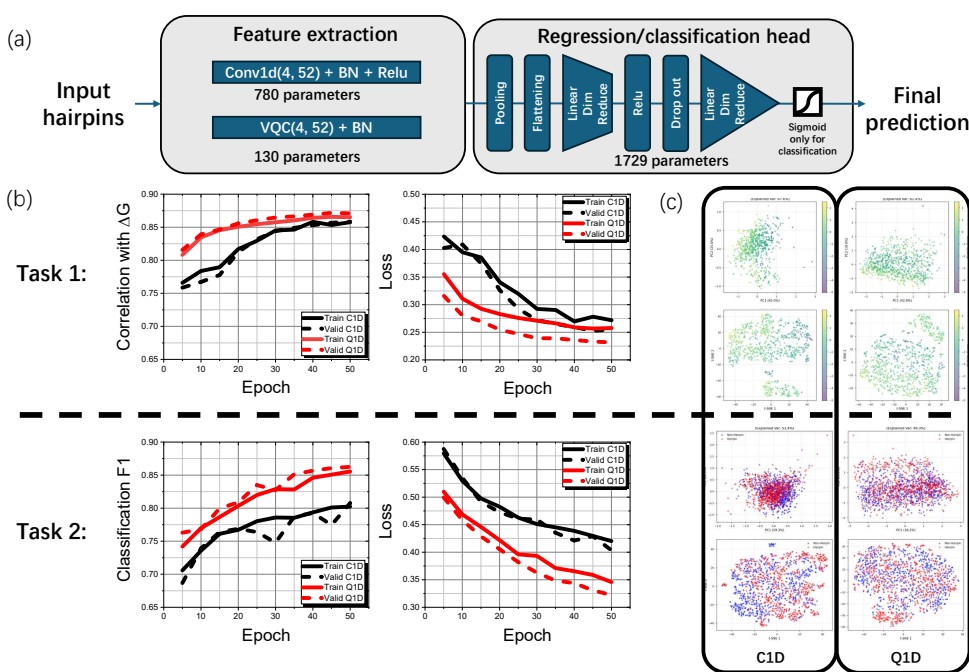

Figure 2: Figure 2: Quantum (Q1D) versus classical (C1D) feature extraction for hairpin motif recognition. (a) Network architecture with Q1D (130 parameters) and C1D (780 parameters) feature extractors feeding into similar heads for hairpin energy regression (Task 1) and hairpin classification (Task 2). (b) Training curves showing Q1D's faster convergence and superior performance in both tasks despite using fewer parameters. (c) PCA and t-SNE visualizations revealing Q1D's enhanced non-linear feature representation with better energy correlation and cluster separation.

Task 1 is to test whether the features can be better correlated with the hairpin energy. Task 2 is to test whether the features can better classify if a certain length of nucleotides could be a hairpin, with additional equal amount of random sequences from the non-hairpin region of different RNAs. Using the same one-hot encoding, the inputs were fed into both the Q1D and the C1D extraction layer. For a fair comparison, C1D used a 4 to 52 channel conv1d with batch norm and relu activation, while Q1D uses quantum circuit with batch norm only. In terms of number of parameters, C1D used 780 parameters and Q1D used 130 parameters. And quantum circuit only used 26 parameters, the additional 104 parameters were attributed to batch norm layer. For both tasks, as shown in figure 2(a), we

used a simple regression head with 1729 parameters. A sigmoid before the classification prediction are added for a yes or no output.

Figure 2(b) and (c) shows the training dynamics, PCA and t-SNE plots comparison between C1D and Q1D respectively. From the training dynamics, it can be seen that Q1D out perform C1D in terms of faster convergence speed, better correlation and higher accuracy in both tasks. For task 1, hairpin energy correlation in reduced space, C1D shows a PCA of -0.041, t-SNE of 0.121, while Q1D shows a PCA of 0.1, t-SNE of 0.331. And also, the first two components PCA explained variance are 68.1% for C1D and 61.4% for Q1D. This is a clear evidence of superior non-linear feature representation of Q1D. Because Q1D's lower PCA explained variance indicates it learns more complex, non-linear features that can't be easily compressed linearly, but these complex features are much better for stability prediction. For task 2, similar results were observed. The first two components PCA explained variance are 51.4% for C1D and 49.3% for Q1D, which would suggest that for this task information is more evenly distributed across all dimensions rather than concentrated in few principal components. Q1D shows slightly better separation with more distinct cluster boundaries in t-SNE clustering. Overall separation of both methods are poor. This might due to additional non-hairpin sequences are similar to hairpin sequences, which makes the classification more challenging. In summary, the Q1D features consistently outperformed C1D features across both hairpin classification and energy correlation, demonstrating superior ability to capture biologically relevant RNA patterns through entanglement-encoded nucleotide correlations, with significantly reduced parameters. To evaluate whether these advantages translate to comprehensive RNA secondary structure prediction, we next conduct benchmark comparisons with SOTA methods.

## 5.2 BENCHMARK COMPARISON WITH EXISTING METHODS

We evaluated Q1Fold performance on the RNAS-trAlign test set, with results summarized in Table 2. Q1Fold achieved competitive performance with an F1 score of 0.963, ranking as the best among all benchmarked methods. Q1Fold demonstrated substantial advantages over traditional energy-based approaches such as RNAfold. When compared to deep learning models like UFold, sincFold, and E2Efold, the performance gap with Q1Fold narrows considerably. The similar performance to sincFold and competitive results with UFold can be attributed to architectural similarities in the second stage of our model, re-

Table 2: Performance comparison on RNAStrAlign test set.

| Method | Precision | Recall | F1 |
|--------|-----------|--------|------|
| **Q1Fold** | 0.972 | 0.958 | 0.963 |
| * UFold | 0.959 | 0.965 | 0.962 |
| DEPfold | 0.948 | 0.974 | 0.960 |
| sincFold | 0.942 | 0.959 | 0.950 |
| E2Efold | 0.649 | 0.789 | 0.705 |
| * RNAfold | 0.515 | 0.568 | 0.539 |

*: reported from original paper or from (WANG & Cohen, 2025)

sulting in comparable performance levels. Notably, Q1Fold outperformed the LLM/FM approach, such as DEPfold. This superior performance is particularly remarkable from a parameter efficiency perspective. Q1Fold requires only 130 parameters for feature generation, while foundation models typically demand billions or trillions of parameters. This positions Q1Fold as both a more accurate and parameter-efficient alternative for RNA 2D structure prediction tasks, when trained on inter-family datasets.

To assessed Q1Fold's generalization ability, we directly test the model trained on the RNAStrAlign training set on ArchiveII dataset, with results summarized in Table 3. Q1Fold achieved a best F1 score of 0.886 on this dateset. Similar to RNAStrAlign dataset, Q1Fold show clear advantages over energy-based models but similar or slightly better performance over deep learning models and LLM/FM models. This demonstrates that Q1Fold has the capacity to generalize to wider range of RNA sequences.

Following prior studies (WANG & Cohen, 2025; Sato et al., 2021), we trained Q1Fold on bpRNA-TR0 and evaluated it on bpRNA-TS0, with results summarized in Table 4. Q1Fold demonstrates competitive performance, achieving the second-best F1 score. On this dataset, Q1Fold's performance fell short of LLM/FM approaches such as DEPfold. This performance gap can be attributed to the substantial difference in feature representation capacity: our quantum circuit generates only 52 feature channels per nucleotide, whereas DEPfold leverages foundation models to produce up to 800 feature channels per nucleotide. Since the bpRNA-1m dataset filters similar structures using an

80% sequence identity cutoff, richer feature representations become crucial for cross-family RNA 2D structure prediction tasks.

Table 3: Performance comparison on ArchiveII test set.

| Method | Precision | Recall | F1 |
|---|---|---|---|
| **Q1Fold** | 0.915 | 0.871 | 0.886 |
| * UFold | 0.876 | 0.890 | 0.881 |
| * RNAErnie | 0.886 | 0.870 | 0.875 |
| DEPfold | 0.852 | 0.820 | 0.830 |
| sincFold | 0.851 | 0.869 | 0.825 |
| * MXfold2 | 0.825 | 0.780 | 0.796 |
| * RNAfold | 0.550 | 0.611 | 0.577 |
| E2Efold | 0.510 | 0.635 | 0.557 |

*: reported from original paper or from (WANG & Cohen, 2025)

Table 4: Performance comparison on bpRNA-TS0 test set.

| Method | Precision | Recall | F1 |
|---|---|---|---|
| DEPfold | 0.686 | 0.636 | 0.644 |
| **Q1Fold** | 0.643 | 0.667 | 0.635 |
| * UFold | 0.587 | 0.711 | 0.630 |
| * RNAErnie | 0.575 | 0.678 | 0.622 |
| sincFold | 0.576 | 0.695 | 0.612 |
| * MXfold2 | 0.519 | 0.646 | 0.558 |
| * RNAfold | 0.446 | 0.631 | 0.508 |
| E2Efold | 0.166 | 0.240 | 0.196 |

*: reported from original paper or from (WANG & Cohen, 2025)

This feature limitation becomes more pronounced on the bpRNA-new dataset, with results summarized in Table 5. As families in the bpRNA-new dataset are not represented in the training set, Q1Fold's performance deteriorates significantly, achieving an F1 score of only 0.406, while traditional thermodynamic models perform reasonably well. Nevertheless, Q1Fold still substantially outperforms early methods like E2Efold, which fails catastrophically on this dataset with an F1 score of 0.051.

Table 5: Performance comparison on bpRNA-new test set.

| Method | Precision | Recall | F1 |
|---|---|---|---|
| * DEPfold | 0.650 | 0.624 | 0.621 |
| * RNAfold | 0.552 | 0.720 | 0.617 |
| **Q1Fold** | 0.544 | 0.352 | 0.406 |
| E2Efold | 0.040 | 0.100 | 0.051 |

*: reported from original paper or from (WANG & Cohen, 2025)

### 5.3 ABLATION

To gain deeper insight into the contribution of different components of Q1D feature extraction layer, we conducted two ablation experiments using bpRNA-TS0 test set. Results summarized in table 6.

**Single-qubit vs multi-qubits measurements** At the measurement stage, we compare the effectiveness of single-qubit measurements only (Q1Fold-SMO) versus multi-qubit correlated measurements only (Q1Fold-MMO). With referring to Table 4, both approaches underperform compared to the combined strategy. Q1Fold-MMO achieves higher recall but lower precision, while Q1Fold-SMO exhibits the opposite pattern—higher precision but lower recall. This indicates that multi-qubit correlated measurements excel at identifying potential base pairs (higher sensitivity), while single-qubit measurements contribute to prediction accuracy (higher specificity). The combination of both measurement strategies achieves optimal recall-precision balance, resulting in superior F1 performance.

Table 6: Ablation study on bpRNA-TS0.

| Method | Precision | Recall | F1 |
|---|---|---|---|
| Q1Fold-SMO | 0.625 | 0.606 | 0.585 |
| Q1Fold-MMO | 0.506 | 0.711 | 0.572 |
| Q1Fold-3qO | 0.567 | 0.686 | 0.600 |

**Biology-aware qubits** We evaluate the impact of biology-aware qubits ($q_3$-$q_5$) by comparing Q1Fold with Q1Fold-3qO, which uses only the three sequence-encoding qubits. The absence of biology-aware qubits results in a 3.5% F1 score reduction, primarily due to decreased precision while recall remains comparable. This demonstrates that encoding positional, contextual, and balance information in dedicated qubits enhances prediction specificity, providing a significant advantage over purely sequence-based encoding. Including biology-aware information in the quantum encoding improves the model's precision without sacrificing recall, demonstrating the value of domain-informed feature design.

## 5.4 LIMITATIONS

Despite the promising results, our approach faces several important limitations that must be acknowledged:

**Computational overhead from simulation.** The most significant practical limitation stems from the current state of quantum computing technology. Due to the limited availability of quantum hardware and the constraints of NISQ devices (noise, limited connectivity, and shallow circuit depth), we rely entirely on classical simulation of quantum circuits. This introduces substantial computational overhead, with the simulation complexity scaling exponentially with the number of qubits. Consequently, Q1Fold's training time is approximately 5–10 times longer than purely classical models when run on GPU-accelerated simulators. This overhead currently negates the theoretical speedup advantages of quantum computing, though this limitation should diminish as quantum hardware becomes more accessible and reliable.

**Limited architectural scalability.** While classical convolutional layers can be easily cascaded to create deep architectures with hierarchical feature extraction, VQCs face fundamental challenges in deep stacking. The cascading of VQCs is not straightforward due to several factors: (1) the measurement collapse that occurs between quantum layers destroys quantum coherence, (2) re-encoding classical outputs back into quantum states introduces additional overhead and potential information loss, and (3) deeper quantum circuits suffer from increased noise accumulation and more severe barren plateau effects. This architectural constraint limits our ability to build deeper quantum networks that might capture more complex hierarchical patterns in RNA structures. Currently, Q1Fold employs only a single quantum layer followed by classical processing, which may limit its capacity to fully exploit quantum advantages for feature learning.

These limitations highlight that while hybrid quantum-classical approaches show promise for RNA structure prediction, significant technological and theoretical advances are still needed to fully realize their potential. Future work should focus on developing more efficient quantum circuit architectures, exploring methods for effective quantum layer stacking, and leveraging emerging quantum hardware as it becomes available.

## 6 CONCLUSION

In this work, we presented Q1Fold, a hybrid quantum-classical convolutional network for RNA 2D structure prediction that demonstrates the practical viability of quantum computing in computational biology. Our approach successfully integrates a 6-qubit VQC for local feature extraction with a classical 2D ResNet architecture for global structure refinement. The experimental results demonstrate that Q1Fold achieves performance comparable to state-of-the-art methods including both traditional energy-based approaches and recent deep learning or language models, while using significantly fewer parameters through quantum feature compression.

The key contribution of our work lies in demonstrating that quantum feature extraction can effectively capture local RNA structural motifs, particularly in hairpin structures where quantum features showed superior correlation with thermodynamic stability scores compared to classical convolutional features. This suggests that quantum entanglement and superposition provide meaningful representational advantages for encoding the complex correlations present in RNA sequences. Furthermore, our qubit-efficient design successfully circumvents the barren plateau problem that has limited previous quantum approaches, making the model trainable on current hardware simulators.

While Q1Fold does not claim superiority over all existing methods, it establishes that hybrid quantum-classical architectures represent a viable and promising direction for RNA structure prediction. The comparable performance achieved with substantially reduced parameter counts suggests potential advantages in scenarios with limited training data, where parameter efficiency becomes crucial for preventing overfitting. As quantum hardware continues to improve, we anticipate that methods like Q1Fold will become increasingly practical and may eventually offer computational advantages beyond parameter efficiency.

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

# A  APPENDIX

## LLM USAGE

We acknowledge the use of Claude (Anthropic) as a general-purpose assistant tool during the preparation of this manuscript. The LLM was utilized in the following capacities: (1) Code development assistance, where Claude helped with code generation for implementing experimental components and data processing scripts, provided debugging suggestions when encountering technical issues, and offered solutions for optimization problems, though all generated code was thoroughly reviewed, tested, and validated by the authors before inclusion in our experimental pipeline; (2) Writing and language polishing, where Claude assisted in improving the clarity and readability of technical descriptions throughout the manuscript, helped refine grammar and sentence structure in various sections, suggested better phrasing and technical terminology to enhance precision, and provided alternatives for complex explanations to improve accessibility for readers; and (3) Quality assurance throughout the writing process, where all LLM-generated content was critically evaluated and verified by the authors for technical accuracy and scientific validity through manual review and cross-checking with relevant literature. We emphasize that the LLM did not contribute to the core research ideas, hypothesis formulation, experimental design, data analysis decisions, or scientific conclusions presented in this work, with its role strictly limited to auxiliary support in implementation and presentation aspects. The authors take full responsibility for all content in this paper, including any portions that were refined with LLM assistance, and confirm that all scientific claims, experimental results, and theoretical contributions are the product of the authors' own research efforts, with the LLM serving only as a tool to enhance the technical execution and written presentation of our original work.

