# OpenReview forum: "Q1Fold: A Qubit-Efficient Hybrid Quantum-Classical Convolutional Neural Network for RNA secondary Structure Prediction"
_ICLR.cc/2026/Conference — ICLR 2026 Conference Withdrawn Submission_

### Official Review · Reviewer_fLkB · 2025-10-30

**Soundness:** 3
**Presentation:** 2
**Contribution:** 2
**Rating:** 4
**Confidence:** 3

**Summary:**

This paper proposes Q1Fold, a hybrid quantum–classical CNN for RNA secondary structure prediction. The model replaces a small portion of the feature extraction pipeline with a 6-qubit variational quantum circuit (VQC) that encodes triplet nucleotide windows (3-mers) into a quantum state. The resulting expectation values (52 observable measurements) are treated as quantum features, which are then processed by a fully classical 2D ResNet CNN to produce RNA contact maps.

The authors argue that this quantum feature extractor provides richer nonlinear correlations than a classical Conv1D while remaining trainable on simulators due to its small qubit count. Experiments on standard datasets (RNAStrAlign, ArchiveII, bpRNA) show competitive performance with state-of-the-art deep learning models, with the added benefit of fewer parameters.

However, the model’s quantum contribution is limited to a single PQC feature extraction layer, and no hardware implementation or theoretical analysis of genuine quantum advantage is provided.

**Strengths:**

* The paper is very well written, with clear figures, diagrams, and ablation studies. The architecture is straightforward, and the authors carefully document how quantum encoding integrates with classical RNA folding pipelines.
* The choice to restrict the circuit to six qubits and to encode meaningful biochemical quantities (position, base type, context, and purine–pyrimidine balance) is thoughtful and well-justified.
* Results on RNA benchmarks are credible and consistent with prior deep learning baselines. The quantum features do not outperform classical ones by a large margin but demonstrate comparable accuracy with fewer parameters.
* Applying hybrid quantum methods to RNA folding, is refreshing and timely, and the results are easier to interpret than toy MNIST-style benchmarks common in quantum ML papers.

**Weaknesses:**

* The quantum component is just a parametric feature map producing classical values for a conventional neural network. The overall system behaves identically to a small nonlinear embedding layer.
* Similar architectures where PQCs act as local filters or “quanvolution” layers already exist in many papers. Q1Fold adapts that idea to RNA data but does not introduce a new computational principle or learning paradigm.
* Motivation of

**Questions:**

* Have you compared the PQC feature extractor to a purely classical nonlinear mapping (e.g., a 1D MLP layer with similar parameter count)?
* Is there any observable property of the quantum circuit (e.g., entanglement entropy or Hilbert-space coverage) correlated with model performance?
* Could your model be implemented on actual NISQ devices, or would noise and limited connectivity immediately destroy its performance?

---

### Official Review · Reviewer_D5rG · 2025-10-31

**Soundness:** 1
**Presentation:** 2
**Contribution:** 1
**Rating:** 0
**Confidence:** 5

**Summary:**

This paper introduces Q1Fold, a hybrid quantum-classical convolutional neural network (HQC-CNN) for RNA secondary structure prediction.
The core innovation is a 6-qubit Variational Quantum Circuit (VQC) that acts as a parameter-efficient 1D feature extractor for local 3-nucleotide windows.
This VQC incorporates a "biology-aware" encoding scheme for positional, contextual, and purine-pyrimidine balance information.
The quantum-derived features are then passed to a classical 2D ResNet, similar to sincFold, to predict the final contact map.
While the authors claim the model is parameter-efficient, its performance is far from the state-of-the-art (SOTA).
The in-family setting (bpRNA-test) is 0.2 lower than deep learning methods like Uni-RNA (2023).
The cross-family setting (bpRNA-test) is 0.2 lower than therodymaic methods like RNAfold (2011).

**Strengths:**

## Parameter Efficiency and Local Feature Power

The model's quantum feature extractor is exceptionally parameter-efficient (130 parameters). The experiments on hairpin motifs provide compelling evidence that the VQC can learn superior representations of local structural features compared to a classical counterpart, which is a valid and interesting finding.

**Weaknesses:**

## Lack of Methodological Novelty

The proposed method appears to be an incremental combination of existing components. The core idea of using a VQC as a convolutional layer is established in prior work, and the classical ResNet backbone is architecturally similar to existing RNA folding networks like sincFold. The work is effectively an "A+B" paper (HQC-CNN + RNA ResNet) and does not offer a fundamental new algorithm or theoretical insight into why quantum features should be uniquely suited for this task beyond parameter efficiency.

## Poor Performance on All Fronts:

The model is not competitive with SOTA.

### Critical Generalization Failure

On the bpRNA-new dataset, which tests for cross-family generalization, Q1Fold's performance collapses (F1 score of 0.406). It is drastically outperformed by older methods like MXfold2 and even the traditional thermodynamic model RNAfold (both >0.6 F1 score), demonstrating a failure to learn generalizable principles of RNA folding.

### Non-Competitive Test Set Performance

Even on the standard bpRNA-TSO test set, the model's performance (F1 0.635) is not SOTA. It is significantly outperformed by more recent methods like BPfold (F1 ~0.701) and large language models such as Uni-RNA and RiNALMo, which have reported much higher scores on this benchmark.

## Unsubstantiated Efficiency Claims and No Practical Advantage

The paper claims parameter efficiency but provides no FLOPs or runtime comparison table to support claims of computational efficiency. In fact, the authors admit their simulation is 5-10 times slower than classical models. Without outperforming classical methods on accuracy or demonstrating a computational speedup, the model offers no practical quantum advantage and is, in its current simulated form, strictly worse than existing classical solutions.

**Questions:**

N/A

**Details Of Ethics Concerns:**

No ethics concerns identified

---

### Official Review · Reviewer_uskz · 2025-10-31

**Soundness:** 3
**Presentation:** 3
**Contribution:** 2
**Rating:** 2
**Confidence:** 3

**Summary:**

The authors propose a quantum computing-based learning framework to tackle the problem of predicting RNA secondary structure from sequence. The main advantage of such approaches lies in the ability to encode high dimensional relationships with few parameters by taking advantage of quantum superposition properties. Evaluation of the method relies on extensive and established benchmarks for RNA secondary structure prediction and performance is strong.

**Strengths:**

* I think this is a highly promising direction of research for the structure prediction field. Of course, current limitations in the QC field are the main bottleneck.
* The paper is well-written and the evaluation is exhaustive.

**Weaknesses:**

* The main weakness in my opinion would be the fit for ICLR. There is no real insight here for the ML community at large.

**Questions:**

* Did you have a look at performance as a function of similarity between train and test sequences?

---

### Official Review · Reviewer_rjLJ · 2025-11-01

**Soundness:** 2
**Presentation:** 2
**Contribution:** 3
**Rating:** 2
**Confidence:** 3

**Summary:**

The paper proposes Q1Fold, a hybrid quantum–classical convolutional architecture for RNA secondary structure prediction. It introduces a 6-qubit variational circuit that operates on local sequence windows to mitigate barren plateau issues, with position-aware, context-aware, and balance-aware quantum encodings informed by RNA biophysics. The quantum features are fed to a lightweight 2D CNN/ResNet head for contact-map prediction. Experiments include multiple standard datasets, an additional hairpin-motif task, and ablations.

**Strengths:**

First attempt (to my knowledge) to embed a hybrid quantum–classical CNN into RNA secondary structure prediction, with a qubit-efficient design.

The position/context/balance-aware qubit mapping reflects domain knowledge and a clear attempt to marry biophysics with quantum feature learning.

Local windowing with only 6 qubits is a sensible strategy to avoid vanishing gradients that often plague VQCs.

**Weaknesses:**

On most datasets Q1Fold is only on par with classical baselines, yet training time is 5–10× longer. On the bpRNA-new cross-family split performance degrades sharply (F1 = 0.406), undermining the generalization claim.

The classic–vs–quantum feature extractor comparison uses ~780 parameters on the classical side vs ~26 for the VQC; such a disparity makes it impossible to attribute gains to “quantumness” rather than simple capacity/regularization differences.

**Questions:**

With matched parameter budgets, how do a classical CNN/conv-MLP and the VQC compare? Please include strict FLOPs/time-budget controlled ablations.

How do you justify “superior non-linear representation” theoretically? Why does lower PCA explained variance imply better structure, rather than added noise? Any information-theoretic or expressivity analyses?

What drives the poor cross-family F1? Does this expose a fundamental limitation of the current quantum feature design (e.g., channel count, encoding bias)? What concrete modifications help (more measurement channels, deeper/shallow-but-wider circuits, alternative encodings)?

---

### Note · Authors · 2025-11-12

I have read and agree with the venue's withdrawal policy on behalf of myself and my co-authors.